# ANYDEPTH: DEPTH ESTIMATION MADE EASY

*"Simplicity is prerequisite for reliability."* — Edsger W. Dijkstra

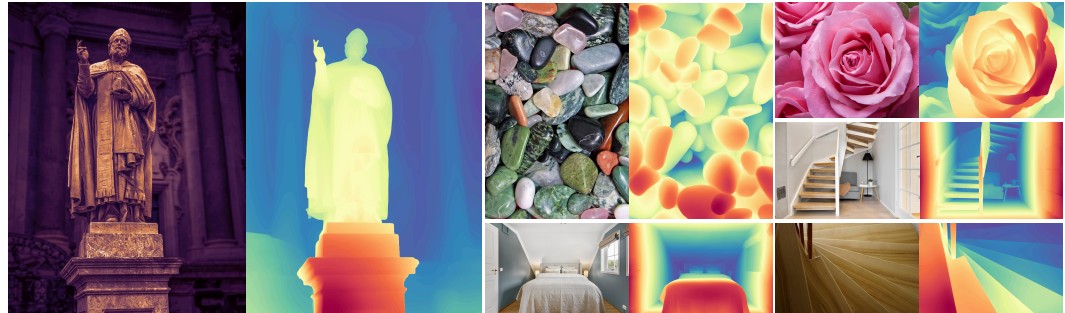

Figure 1: We present **AnyDepth**, a simple and efficient training framework for zero-shot monocular depth estimation, which achieves impressive performance across a variety of indoor and outdoor scenes.

## ABSTRACT

Recent monocular depth estimation models have achieved impressive performance. However, they typically rely on traditional encoders, complex decoders, and large training datasets, which collectively limit their efficiency and generalization. In this work, we pursue a complementary approach: building a lightweight and efficient training framework without sacrificing accuracy. First, we apply DI-NOv3 to zero-shot monocular depth estimation for the first time. Second, we design a lightweight decoder that reduces parameters and computational cost while maintaining competitive performance. Third, inspired by data-centric learning, we propose a quality-based filtering strategy to filter out harmful samples, thereby reducing dataset size while improving overall training quality. Experiments on multiple benchmarks demonstrate that our approach achieves comparable or even superior accuracy to larger-scale counterparts, despite using fewer parameters and data. Our work emphasizes the integration of visual backbone performance, decoder efficiency, and data quality to explore more efficient zero-shot monocular depth estimation pipelines.

## 1 INTRODUCTION

Monocular depth estimation is gaining increasing attention due to its wide range of downstream applications. Depth maps are not only used to measure scene distances (Bhat et al., 2023; 2021; Godard et al., 2017), but can also be embedded as conditional information within models in the 3D generation domain (Zhang et al., 2023; Rombach et al., 2022; Poole et al., 2022; Mildenhall et al., 2021; Li et al., 2024a; Yang et al., 2023), providing complementary information to improve granularity and geometric consistency.

The MiDaS series (Ranftl et al., 2020; Birkl et al., 2023), through extensive and systematic experiments, compared the transfer performance of various pretrained vision transformers (such as ViT (Dosovitskiy et al., 2020), Swin (Liu et al., 2021), DINO (Oquab et al., 2023), and BeiT (Bao et al., 2021)) on monocular depth estimation tasks. These experiments demonstrated the crucial importance of semantic information extracted by pretrained models for improving model performance.

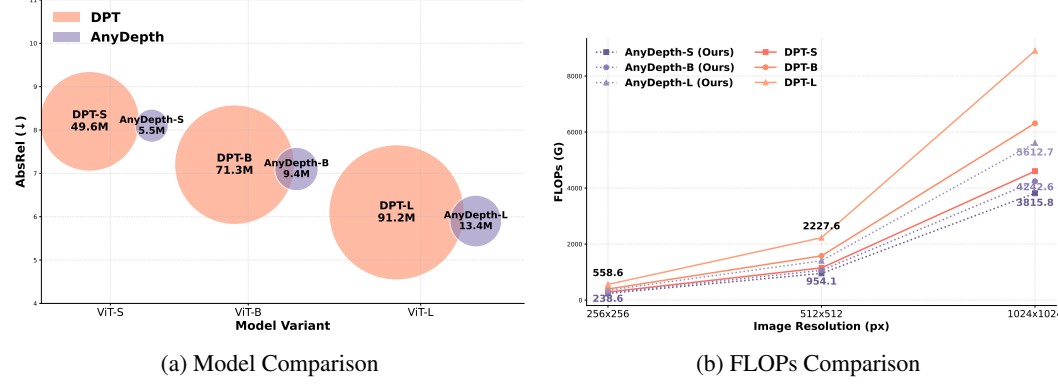

(a) Model Comparison (b) FLOPs Comparison

Figure 2: Comparison of the number of parameters (left) and computational complexity (right) of **AnyDepth** and DPT for different model sizes and input resolutions. Our method significantly reduces the number of model parameters and computational cost while maintaining competitive accuracy.

High-quality dense feature maps are required to obtain finer-grained depth maps, while global context and long-range dependency modeling are essential to preserve geometric structure.

DPT (Ranftl et al., 2021) has demonstrated impressive performance in various dense prediction tasks and is currently used as the decoder in mainstream models. DPT aims to achieve finer-grained predictions by fusing features at different scales. However, each Transformer layer requires a separate Reassemble module to map to different scales, followed by multiple alignments. This results in a cumbersome architecture and high computational overhead. Second, this structure also leads to large parameters and slow inference speed.

The Depth Anything series (Yang et al., 2024a;b) represents a typical data-driven approach, aiming to improve understanding and generalization capabilities of model for complex scenarios by leveraging massive datasets. These methods have significantly improved performance in zero-shot scenarios, demonstrating the potential of data scalability in the field of depth estimation. However, this purely data-driven approach also has drawbacks. First, data collection is expensive, hindering academic replication and research. Second, large datasets inevitably contain noisy samples, which can negatively impact model training. Therefore, simply using data-driven approaches to improve model performance is limited.

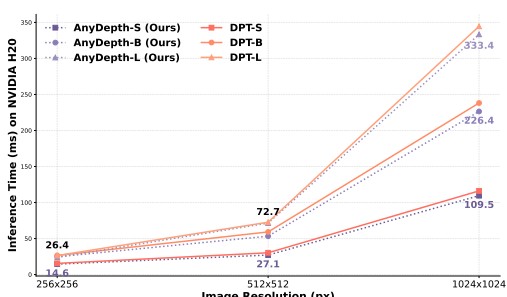

Figure 3: Comparison of inference time between AnyDepth and DPT at different input resolutions. Our method consistently achieves lower latency, especially at higher resolutions.

Based on these findings and limitations, we re-examined the entire monocular depth estimation pipeline, aiming to design a lightweight and efficient training framework that maintains competitive performance while being widely adopted by the research community (Fig. 2).

Specifically, our contributions are reflected in four aspects:

- We leverage the latest DINOv3 encoder to provide higher-quality dense features for depth estimation. To our best knowledge, this is the first application of DINOv3 to a monocular depth estimation framework.
- We design a novel decoder that aligns and fuses features before restoring resolution through a one-shot reconstruction and upsampling. This architecture avoids multi-branch cross-scale alignment and repeated reconstruction, better preserving high-frequency details and geometric consistency.
- We analyze sample quality issues in deep learning datasets and proposed two metrics to quickly measure sample quality, which we then used to filter out low-quality samples. This

reduced dataset size while improving overall data quality, demonstrating that our framework can achieve better performance with fewer resources.

- On multiple benchmarks, our framework achieves comparable accuracy and generalization to DPT with significantly fewer parameters and lower training overhead, demonstrating a superior efficiency-accuracy trade-off and academic reproducibility.

## 2 RELATED WORK

**Zero-Shot Monocular Depth Estimation.** To enable widespread use of depth images in real-world scenarios without relying on specific environments, zero-shot depth estimation has become a key research direction in recent years (Chen et al., 2016; Piccinelli et al., 2024; Chen et al., 2020; Yin et al., 2021). Due to the lack of strict geometric constraints on MDE, many zero-shot models learn to predict affine-invariant depth, i.e., recovering relative structure while maintaining scale and translation invariance (Ranftl et al., 2020; Yang et al., 2024a;b). For example, DiverseDepth (Yin et al., 2020) uses web images as training data to improve zero-shot generalization performance. MiDaS (Ranftl et al., 2020) proposed scale-shift-invariant losses to solve the ambiguity problem of different deep numerical representation methods of different datasets, so that the model can be trained on a large scale. In order to eliminate the inherent problems of the CNN backbone, the performance of Zero-Shot Monocular Depth Estimation was further improved by using the vision transformer architecture, such as DPT (Ranftl et al., 2021), Omnidata (Eftekhar et al., 2021), Depthformer (Li et al., 2023) and Zoepdeth (Bhat et al., 2023). Marigold (Ke et al., 2024) directly utilizes the standard diffusion model paradigm and stable diffusion pre-trained weights for fine-tuning to produce high-quality results. Depth Anything series (Yang et al., 2024a;b) used 62 million unlabeled images for larger-scale training. Geowizard (Fu et al., 2024) uses the high consistency between dense prediction tasks to jointly predict depth and normals. Lotus (He et al., 2024) analyzes the diffusion process to achieve single-step diffusion and speed up the inference process. Genpercept (Xu et al., 2024) uses experiments to prove that the diffusion model requires specific details to be optimized in dense prediction tasks.

**Decoder for Dense Prediction.** Currently, many methods for dense prediction tasks employ multi-scale feature fusion strategies to compensate for the lack of information from single-layer features (Lin et al., 2017; Liu et al., 2018; Tan et al., 2020; Chen et al., 2018; Ghiasi et al., 2019; Xu et al., 2021; Eigen & Fergus, 2015). FPN (Lin et al., 2017) proposes a top-down architecture where high-level semantic representations are successively merged with low-level features to enhance multi-scale features. (Lee et al., 2019) designed a multi-scale local plane guidance layer to more effectively guide the fusion of features at each layer to achieve performance improvement. Swin-Depth (Cheng et al., 2021) designs a lightweight multi-scale attention mechanism module to enhance the ability to learn global information at multiple scales. PVT (Wang et al., 2021) and Uformer (Wang et al., 2022) use a multi-scale pyramid decoder structure to capture long-range visual dependencies.DPT (Ranftl et al., 2021) utilizes the ViT (Dosovitskiy et al., 2020) backbone network to generate high-resolution features, thereby achieving finer-grained representation and improving prediction accuracy. However, multi-branch reassembly incurs significant overhead, especially in the case of high-resolution input.

## 3 THE PROPOSED METHOD

### 3.1 OVERVIEW

The proposed AnyDepth uses a pre-trained DINOv3 (Siméoni et al., 2025) encoder and SDT decoder; as shown in Fig.4, given an input image $I$, we extract multi-scale representations from four intermediate Transformer layers $T^1, T^2, T^3, T^4$ and input them into the SDT head for depth reconstruction, thereby capturing different levels of detail and semantic information. These tokens are linearly projected onto a common dimension and fused to capture complementary semantic levels. The fused representations are then reshaped into feature maps and refined by a Spatial Detail Enhancer (SDE). Finally, a dense depth map is generated through two learnable Upsampler and head prediction.Our method differs from the Depth Anything series (Yang et al., 2024a;b) and DPT (Ranftl et al., 2021) in that we fuse tokens using only a single linear projection, followed by upsampling

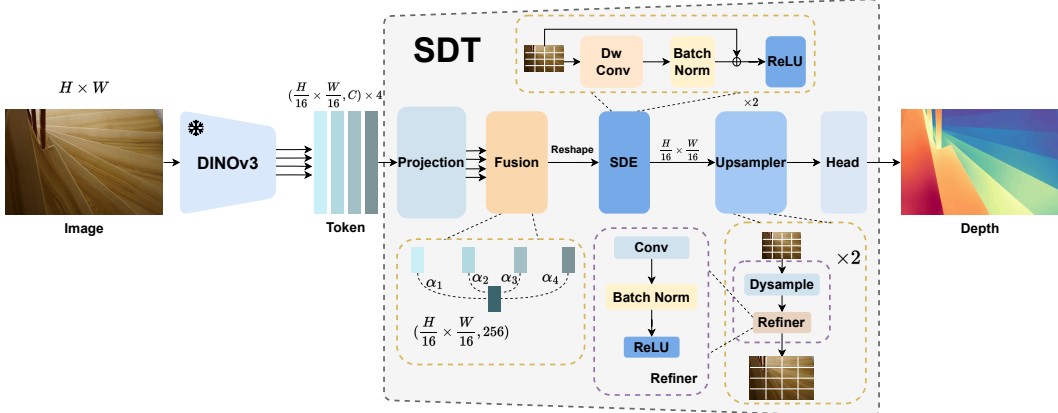

Figure 4: **AnyDepth architecture overview.** The input image is encoded into tokens by a frozen DINOv3 backbone network, then decoded by our lightweight SDT decoder. Tokens undergo only a single projection and weighted fusion. The Spatial Detail Enhancer (SDE) module ensures finer-grained predictions. The feature map is upsampled by an efficient and learnable upsampler dysample, and the depth is finally output by the head.

in a single path, without multi-branch cross-scale alignment, significantly reducing the number of parameters and computational overhead.

## 3.2 SIMPLE DEPTH TRANSFORMER (SDT)

Our decoder adopts a simple single-path fusion and reconstruction strategy, aiming to take advantage of the high-resolution feature of DINOv3 and further unleash its performance at high resolution. We first project the tokens extracted from the encoder into a 256-dimensional space using a linear layer followed by a GELU non-linearity (Hendrycks & Gimpel, 2016), which preserves sufficient informative content while substantially reducing the computational overhead in the subsequent decoding stages. For the class token, we keep the same processing as DPT (Ranftl et al., 2021), concatenate it with the spatial token, and then fuse it through the learnable projection.

**Fusion.** To fuse tokens from multiple layers of representation, we then employ a learnable weighted fusion strategy (Eq. 1).

Specifically, we assign a learnable scalar weight to each layer of tokens and normalize them using a softmax function to form a uniform probability distribution, preventing initial instability in training. This strategy enables the model to adaptively balance low-level structural details with high-level semantic information.

$$T = \sum_{i \in \mathcal{L}} \alpha_i \operatorname{Proj}_i(T_i), \quad T_i \in \mathbb{R}^{N_P \times D}, \tag{1}$$

Where $T_i$ denotes the token in layer $i$ after projection, and contains $N_p$ tokens of dimension $D$.

**Spatial Detail Enhancer.** After the fusion block, we reshape the sequence token output into a spatial feature map. Because the reorganized feature map lacks local continuity and, after multi-level fusion, easily obscures shallow texture details, which are crucial for dense prediction tasks such as depth estimation, we designed the Spatial Detail Enhancer.The SDE can be expressed by Eq. 2,

$$F' = ReLU(F + BN(DWConv_{3 \times 3}(F))), \ F \in \mathbb{R}^{\frac{H}{16} \times \frac{W}{16} \times 256}. \tag{2}$$

We implement this operation first using a $3 \times 3$ Depthwise convolution for local spatial modeling, followed by batch normalization. We then add the normalized response to the input feature $F$ via a residual connection, and finally pass it through an activation layer.

**Upsampler.** In the upsampling stage, we abandon the commonly used bilinear interpolation, which easily blurs high-frequency details, and instead adopt a learnable dynamic sampler (Eq. 6). Specifically, we use DySample (Liu et al., 2023) as the upsampler, which adaptively constructs an offset sampling grid based on the learned low-resolution features to adjust the sampling position,

and then uses differentiable grid sampling to resample to high-resolution features. We first define three operators: the DySample block $\mathcal{B}(\cdot)$, the DySample stage $\mathcal{S}(\cdot)$, and the refinement block $\mathcal{R}(\cdot)$:

$$\mathcal{B}(X) = \text{ReLU}\Big(\text{BN}\big(\text{Conv}_{3\times3}(\text{DySample}_{\times2}(X))\big)\Big), \tag{3}$$

$$\mathcal{S}(X) = \mathcal{B}\big(\mathcal{B}(X)\big), \tag{4}$$

$$\mathcal{R}(X) = \text{ReLU}\Big(\text{BN}\big(\text{Conv}_{3\times3}(X)\big)\Big). \tag{5}$$

Based on these definitions (Eq. 3, 4, 5), the complete upsampling process can be expressed as:

$$\mathcal{U}(X) = \mathcal{R}\Big(\mathcal{S}\big(\mathcal{R}(\mathcal{S}(X))\big)\Big), \tag{6}$$

In this way, the compact feature map of size $H/16 \times W/16$ can be progressively upsampled back to the original resolution $H \times W$. We want to emphasize that we do not jump to $H \times W$ all at once, but rather decompose the upsampling into two $\times4$ upsamplers, using four dysamples of scale 2. Single-stage $\times16$ upsampling forces the sampler to infer large offsets from very low-resolution features, which amplifies errors and destabilizes gradients. Our progressive design keeps the offsets small, inserting local refinement after each resampling, resulting in a model with better detail recovery capabilities.

## 3.3 SDT VS. DPT

A key difference between SDT and DPT (Ranftl et al., 2021) is the order of feature reassembly. DPT employs a reassemble-fusion strategy. Specifically, DPT first applies the reassemble module to the tokens extracted by each Transformer layer, mapping the tokens to feature maps of different scales. These feature maps are then fused in a cascade across scales, which inevitably introduces multiple branches and repeated cross-scale alignment overhead. In contrast, SDT employs a fusion-reassemble strategy, directly projecting and fusing groups of tokens. Only after this stage do we perform spatial reassembly and upsampling along a single path. This fusion-reassemble strategy avoids the high cost of per-layer token reassembly and feature map cross-scale alignment, making it more efficient and stable, especially when processing high-resolution inputs.

## 3.4 DATA CENTRIC LEARNING

Although MiDaS (Ranftl et al., 2020) uses an affine-invariant loss to accommodate multi-dataset training, the varying degrees of noise and scale ambiguity introduced by these datasets can easily negatively impact training, especially in dense prediction tasks (Fig.7, 8). Inspired by data-centric learning (Singh, 2023; Zha et al., 2025), for the monocular depth estimation task and our setting, we believe that high-quality samples should possess two properties: (i) depth values should be evenly distributed throughout the image, rather than being overly concentrated within a specific range; and (ii) gradient magnitudes should vary slightly across continuous surfaces, while exhibiting more pronounced changes near object edges. Based on these two properties, we define two metrics to measure sample quality. These metrics aim to reduce low-quality samples, facilitate model training, and reduce dataset size and training cost.

### 3.4.1 DEPTH DISTRIBUTION SCORE

Some samples have depths that are primarily concentrated near or far, while other depth ranges are relatively small. As shown in Fig. 7 , this phenomenon is common in outdoor datasets. This unbalanced depth distribution can cause the model to favor learning depth values within a specific range rather than the entire valid depth range, leading to unstable training and poor model generalization.

To quantify this phenomenon, we propose a *Depth Distribution Score* that evaluates how uniformly depth values are distributed across the available depth range. For a depth map $D \in \mathbb{R}^{H \times W}$, we divide the depth values into $K$ bins of equal width, and we use $K = 20$ by default to balance granularity and robustness.

**Chi-square Deviation** ($S_{\chi^2}$). We measure the deviation from a uniform distribution using the chi-square statistic:

$$\chi^2 = \sum_{k=1}^{K} \frac{(n_k - \bar{n})^2}{\bar{n}}, \quad S_{\chi^2} = \exp\left(-\frac{\chi^2}{N}\right), \tag{7}$$

where $n_k$ is the number of depth bins $k$, $\bar{n} = N/K$ is the expected number under a uniform distribution, and $N$ is the total number of valid depth values. We use an exponential transformation to map the chi-squared statistic (Eq. 7) to $[0, 1]$, with higher scores indicating a more uniform distribution.

**Maximum Concentration Index** ($S_{\mathrm{conc}}$). To prevent excessive concentration in any single depth interval, we penalize the maximum bin occupancy:

$$S_{\mathrm{conc}} = \begin{cases} 1, & \text{if } p_{\max} \leq 2/K \\ 1 - \min\left(1, \frac{p_{\max} - 2/K}{0.5 - 2/K}\right), & \text{otherwise} \end{cases} \tag{8}$$

where $p_{\max} = \max_k(n_k)/N$ is the maximum bin probability. This formulation (Eq. 8) tolerates up to twice the ideal concentration ($2/K$) without penalty, then linearly decreases the score as concentration increases.

**Range Utilization** ($S_{\mathrm{range}}$) . Partition the available depth range into $K$ equal-width bins and let $n_k$ be the count in bin $k$. Define the number of non-empty bins $K_+ = \{\, k \in \{1, \ldots, K\} \mid n_k > 0 \,\}$. The range utilization score is $S_{\mathrm{range}} = K_+/K$, which penalizes samples whose depths concentrate within a narrow portion of the range.

The final Depth Distribution Score $S_{\mathrm{dist}}$ is the weighted sum of these three scores:

$$S_{\mathrm{dist}} = \lambda_1 \cdot S_{\chi^2} + \lambda_2 \cdot S_{\mathrm{conc}} + \lambda_3 \cdot S_{\mathrm{range}}, \tag{9}$$

where we empirically set $\lambda_1 = 0.5$, $\lambda_2 = 0.3$, and $\lambda_3 = 0.2$.

### 3.4.2 GRADIENT CONTINUITY SCORE

In the real world, continuous physical surfaces should have smoothly transitioning depth values, without drastic random fluctuations. However, perhaps due to rendering defects in synthetic data, some sample depth maps exhibit gradient abrupt changes caused by noise on smooth surfaces. If these samples are used for training, the model will learn incorrect depth changes, thus affecting prediction quality.

Inspired by the gradient loss function ((Li et al., 2024b; Yang et al., 2018; Ranftl et al., 2020)), we propose a *gradient continuity score* to assess the noise content of each sample. We first calculate the gradient magnitude $G(i, j) = \sqrt{(\partial_x D)^2 + (\partial_y D)^2}$. To distinguish reasonable gradient abrupt changes at normal object edges from those caused by abnormal noise, we define edge pixels as pixels with gradient magnitudes in the top $10\%$. Within the smooth region, we use the coefficient of variation $\mathrm{CV} = \frac{\sigma_G}{\mu_G}$ to assess gradient consistency:

$$S_{\mathrm{grad}} = \frac{1}{1 + \mathrm{CV}}, \tag{10}$$

where $\mu_G$ and $\sigma_G$ are the mean and standard deviation of the gradient magnitude in the region, respectively.

### 3.4.3 TOTAL SCORE

The depth distribution score and gradient continuity score capture different aspects of sample quality. We combine them into a *Total Score*, defined as $S_{\mathrm{total}} = (S_{\mathrm{grad}} + S_{\mathrm{dist}})/2$, to assess the overall quality of each sample for dataset filtering (Eq. 9, 10). It's important to note that our goal is not to provide a particularly precise quality assessment method, but rather to design efficient indicators to quickly filter out samples with quality issues. For example, when performing edge detection, we did not use traditional Canny or Sobel algorithms because the detected edge maps often produce unnecessary artifacts and details. Learning-based methods, on the other hand, predict edges that are always several pixels off from their exact locations (Li et al., 2024b; He et al., 2019; Pu et al., 2022; Su et al., 2021), and their inference time is time-consuming, making them unsuitable for rapid filtering of large datasets.

## 4 EXPERIMENTS

### 4.1 DATASETS AND METRICS

**Training Datasets.** We use five synthetic datasets covering various indoor and outdoor scenes for training. (1) *Hypersim* (Roberts et al., 2021) after filtering incomplete samples, we have approximately 39K. (2) *Virtual KITTI* (Cabon et al., 2020) we selected four scenes, totaling approximately 20K. (3) *BlendedMVS* (Yao et al., 2020) (4) *IRS* (Wang et al., 2019) (5) *TartanAir* (Wang et al.) As shown in Table 1, we only use 369K datasets for training. The far plane is set to $100\,\mathrm{m}$. To improve the robustness and generalization of the model, we used data augmentation of flipping and rotation.

**Evaluation Datasets and Metrics.** For Zero-shot monocular depth estimation, we evaluate SDT using five datasets containing various scenes: NYUv2 (Silberman et al., 2012), KITTI (Geiger et al., 2013), ETH3D (Schops et al., 2017), ScanNet (Dai et al., 2017), and DIODE (Vasiljevic et al., 2019). We use the absolute mean relative error(AbsRel), i.e., $\frac{1}{M}\sum_{i=1}^{M}\frac{|\hat{d}_i-d_i|}{d_i}$, where $M$ is the total number of valid pixels, $d_i$ denotes the ground truth, and $\hat{d}_i$ is the predicted depth. We report accuracy thresholds $\delta_\tau$, which denote the fraction of pixels where the prediction and ground truth differ by less than a multiplicative factor $\tau = 1.25$.

### 4.2 IMPLEMENTATION DETAILS

Our setup differs slightly from Depth Anything V2 (Yang et al., 2024b). To better utilize the high-resolution features of DINOv3 (Siméoni et al., 2025), we increase the input image resolution to $768 \times 768$. The encoder is kept frozen throughout training, and we use features from four intermediate layers as decoder inputs: $[2, 5, 8, 11]$ for DINOv3 S/16 and DINOv3 B/16, and $[4, 11, 17, 23]$ for DINOv3 L/16. We perform simple regression to predict disparity $d' = 1/d$, where $d'$ denotes disparity and $d$ denotes depth. Both the input image and the groundtruth are normalized to $[0, 1]$. We follow the settings of Depth Anything v2 (Yang et al., 2024b) and use a scale- and shift-invariant loss $\mathcal{L}_{\mathrm{ssi}}$ and a gradient matching loss $\mathcal{L}_{\mathrm{gm}}$, and the weight ratio of $\mathcal{L}_{\mathrm{ssi}}$ and $\mathcal{L}_{\mathrm{gm}}$ is set to $1:2$. To stabilize optimization, we follow an optimization strategy similar to DINOv3 (Siméoni et al., 2025). We use AdamW with a base learning rate of $1 \times 10^{-3}$, a PolyLR scheduler with power $0.9$, and a linear warm-up for the first two epochs. We train for a total of five epochs.

### 4.3 MAIN RESULTS

#### 4.3.1 RESULTS OF DATA CENTRIC LEARNING

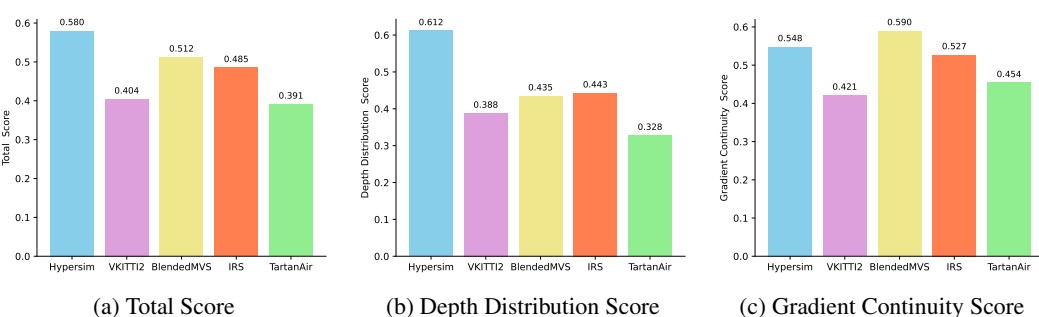

(a) Total Score      (b) Depth Distribution Score      (c) Gradient Continuity Score

Figure 5: Dataset quality across the Total Score, Depth Distribution Score, and Gradient Continuity Score (higher is better).

We applied the metrics proposed in Section 3.4 to all training datasets, with the results shown in Fig. 5. We observe that Hypersim performed well in both the Depth Distribution Score and Gradient Continuity Score, achieving the highest overall score. This indicates a relatively balanced depth distribution, smooth gradients, and a low concentration of noisy samples. In contrast, datasets containing outdoor samples, such as VKITTI2, BlendedMVS, and TartanAir, had significantly lower Depth Distribution Scores, indicating a more severe depth distribution. This is likely a common problem across all outdoor datasets. The low Gradient Continuity Score for VKITTI2 may be due to

Table 1: Dataset statistics of good and bad samples.

| Dataset | Total | Good | Bad |
|---|---|---|---|
| Hypersim | 39,648 | 26,912 | 12,736 |
| VKITTI2 | 19,559 | 12,643 | 6,916 |
| BlendedMVS | 115,142 | 74,838 | 40,304 |
| IRS | 103,316 | 68,211 | 35,105 |
| TartanAir | 306,637 | 186,693 | 119,944 |
| **Summary** | 584,302 | 369,297 | 215,005 |

Table 2: Quantitative comparison of zero-shot affine-invariant depth estimation. Lower AbsRel values are better; higher $\delta_1$ values are better. DINOv3 (Siméoni et al., 2025) uses the ViT-7B encoder, and Depth Anything v2 (DAv2) (Yang et al., 2024b) is trained on 62.6M datasets. For fair comparison, the baseline (DPT) uses a frozen DINOv3 encoder and DPT head, while our method replaces the DPT head with the proposed SDT. The bold numbers in the table refer to the best results between DPT and AnyDepth.

| Method | Training Data↓ | Encoder | #Params (M)↓ | NYUv2 AbsRel↓ | $\delta_1$ ↑ | KITTI AbsRel↓ | $\delta_1$ ↑ | ETH3D AbsRel↓ | $\delta_1$ ↑ | ScanNet AbsRel↓ | $\delta_1$ ↑ | DIODE AbsRel↓ | $\delta_1$ ↑ |
|---|---|---|---|---|---|---|---|---|---|---|---|---|---|
| DINOv3 | 595K | ViT-7B | 91.19 | 4.3 | 98.0 | 7.3 | 96.7 | 5.4 | 97.5 | 4.4 | 98.1 | 25.6 | 82.2 |
| DAv2 | 62.6M | ViT-S | 70.64 | 5.3 | 97.3 | 7.8 | 93.6 | 14.2 | 85.1 | – | – | 7.3 | 94.2 |
|  |  | ViT-B | 157.33 | 4.9 | 97.6 | 7.8 | 93.9 | 13.7 | 85.8 | – | – | 6.8 | 95.0 |
|  |  | ViT-L | 391.19 | 4.5 | 97.9 | 7.4 | 94.6 | 13.1 | 86.5 | – | – | 6.6 | 95.2 |
| DPT | 584K | ViT-S | 49.64 | 8.4 | 93.3 | 10.8 | 89.1 | 12.7 | 92.0 | 8.3 | 93.5 | 26.0 | 71.4 |
|  |  | ViT-B | 71.33 | 7.5 | 95.1 | 10.8 | 88.9 | 10.0 | 92.9 | 7.1 | 95.3 | 24.5 | 73.4 |
|  |  | ViT-L | 91.19 | 6.1 | **96.8** | 8.9 | 92.5 | 13.0 | 94.9 | 6.0 | 97.0 | 23.4 | **73.9** |
| **AnyDepth** | 369K | ViT-S | 5.52 | 8.2 | 93.2 | 10.2 | 88.3 | 8.4 | 93.5 | 8.0 | 93.6 | 24.7 | 71.4 |
|  |  | ViT-B | 9.45 | 7.2 | 95.0 | 9.7 | 90.1 | **8.0** | 94.5 | 6.8 | 95.6 | 23.6 | 72.7 |
|  |  | ViT-L | 13.38 | **6.0** | 96.8 | **8.6** | **92.6** | 9.6 | **95.4** | 5.4 | **97.4** | **22.6** | 73.6 |

the presence of numerous fine-grained structures (*e.g.*, leaves) in the samples, resulting in abundant edges and severe gradient abruptness, which is considered noisy.

Following the methods described in Section 3.4, we filtered the entire dataset. Specifically, we first filtered out samples whose valid depth values accounted for less than 20% of the total pixels. We then sorted the remaining samples based on the Depth Distribution Score and Gradient Continuity Score, filtering out the 20% with the lowest scores for each metric. The number of filtered samples for each dataset is shown in Table 1. For visualizations of low-quality samples, please see the A.2. The merged dataset contains 584K samples, of which approximately 369K are used for training and 215K are filtered out.

### 4.3.2 QUANTITATIVE COMPARISONS

Table 2 reports quantitative comparison results for zero-shot affine-invariant depth estimation. Since the baselines in the Depth Anything series all use a DPT head, we primarily compare our proposed SDT decoder with the DPT under the same backbone settings.

While our approach does not yet surpass the state-of-the-art results reported by fully data-driven methods (*e.g.*, the Depth Anything series (Yang et al., 2024a;b) and DINOv3-7B (Siméoni et al., 2025), which require hundreds of millions of parameters or massive datasets), we emphasize that our entire AnyDepth is designed from a light-weight and simple perspective, focusing not only on model design but also on data quality and quantity. Inspired by the principles of data-centric learning, we conclude that our model can achieve superior performance even with a relatively small amount of high-quality data (369K).

SDT uses only 5–13M parameters and outperforms DPT with various encoder sizes. Our results show that SDT significantly reduces the number of parameters and training cost while maintaining comparable accuracy to DPT, and there is a slight improvement in inference speed (Fig. 3). AnyDepth provides a lightweight, efficient, and computationally friendly alternative.

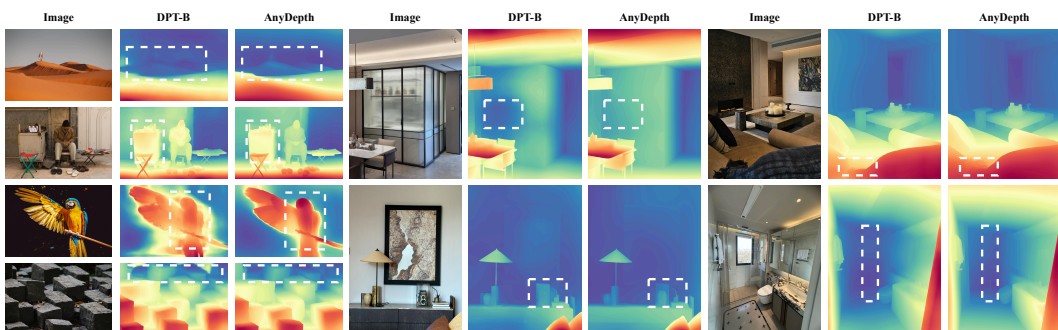

Figure 6: Qualitative results of zero-shot monocular depth estimation using **AnyDepth** of ViT-B and comparison with DPT-B.

Table 3: Ablation on the effect of data filtering across five benchmarks. Filtering denotes our quality-based sample selection strategy (369K vs. 584K). We report AbsRel (lower is better) and $\delta_1$ (higher is better).

| Variant | NYUv2 | | KITTI | | ETH3D | | ScanNet | | DIODE | |
|---|---|---|---|---|---|---|---|---|---|---|
| | AbsRel↓ | $\delta_1$ ↑ | AbsRel↓ | $\delta_1$ ↑ | AbsRel↓ | $\delta_1$ ↑ | AbsRel↓ | $\delta_1$ ↑ | AbsRel↓ | $\delta_1$ ↑ |
| w/o Filtering | 7.3 | **95.1** | **9.3** | **91.1** | 8.8 | **94.8** | 7.0 | **95.6** | 24.4 | **72.7** |
| **w/ Filtering (ours)** | **7.2** | 95.0 | 9.7 | 90.1 | **8.0** | 94.5 | **6.8** | **95.6** | **23.6** | **72.7** |

## 4.4 EFFICIENCY

We comprehensively evaluated efficiency advantages of AnyDepth. Compared to DPT, AnyDepth not only significantly reduces the number of parameters (Fig.2a), but also shows that AnyDepth significantly reduces FLOPs by 37% when using models of varying sizes, particularly at high resolutions (Fig.2b). It also slightly improves inference speed (Fig.3). Furthermore, Average iteration time of AnyDepth during training is 10% shorter than that of DPT.

## 4.5 ABLATION STUDY

To examine the effectiveness of our data-centric filtering strategy, we compared models trained using all available samples (584K) with models trained using a filtered subset (369K). As shown in Table 3 , while the amount of training data is reduced, model performance remains comparable. This suggests that removing noisy or low-quality samples benefits model training, further supporting our data-centric approach.

## 5 LIMITATIONS AND FUTURE WORK

While our work demonstrates advantages, it also has some limitations. First, the current pipeline has not been evaluated in large-scale fully supervised or fine-tuned settings. Second, further analysis of the dataset can be used to optimize the filtering strategy. In future work, we can extend our lightweight framework to a wider range of tasks, such as metric depth and normal estimation.

## 6 CONCLUSION

In this paper, we introduce AnyDepth, a simple and efficient-to-train framework for zero-shot monocular depth estimation. In our setup, a powerful self-supervised visual backbone paired with a single-path lightweight decoder is sufficient to achieve competitive performance without the need for large-scale, costly training. The goal of AnyDepth is not to surpass large-scale state-of-the-art methods, but rather to provide a more practical and academically valuable approach through its lightweight design and improved data quality.

ETHICS STATEMENT

This research focuses on monocular depth estimation using publicly available synthetic and real-world datasets. All datasets are widely used in the literature and were collected with relevant consent and licenses. No new human or animal data were collected for this research. Our methods do not involve personally identifiable information and do not allow for surveillance or privacy-intrusive applications outside of existing depth estimation research. We release our code and data processing pipeline under an open license to support transparency and research integrity. This research adheres to the ICLR ethical guidelines on fairness, accountability, and responsible dataset use.

REPRODUCIBILITY STATEMENT

We have made every effort to ensure the reproducibility of our results. To further enhance reproducibility, we will release anonymized source code, pretrained model checkpoints, and data filtering scripts. All datasets used in this research are publicly available. These resources enable independent researchers to replicate our experiments and verify all reported results.

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

# A APPENDIX

## A.1 LLM USE DECLARATION

Large Language Models (ChatGPT) were used exclusively to improve the clarity and fluency of English writing. They were not involved in research ideation, experimental design, data analysis, or interpretation. The authors take full responsibility for all content.

## A.2 VISUALIZATION OF LOW-QUALITY SAMPLES

Figure 7 provides qualitative examples of low-quality samples from five training datasets. It can be seen that some datasets contain samples with highly uneven depth value distributions, leading to biased supervision. This situation motivates us to use a depth distribution score when evaluating dataset quality.

In addition, Figure 8 shows RGB images, gradient maps, and ground-truth depth examples from the same five datasets. The highlighted areas indicate the presence of severe gradient noise or inconsistent edges, which can negatively impact training stability. These qualitative findings support our quantitative gradient consistency metric.

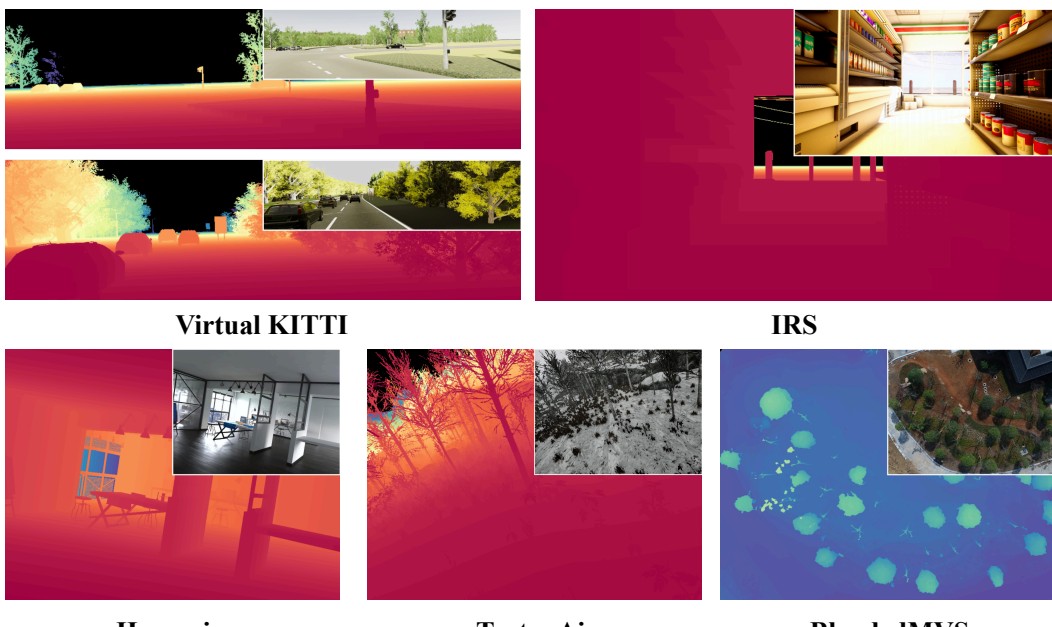

Figure 7: RGB images and GT of each dataset, showing that the depth value distribution of some samples is not uniform.

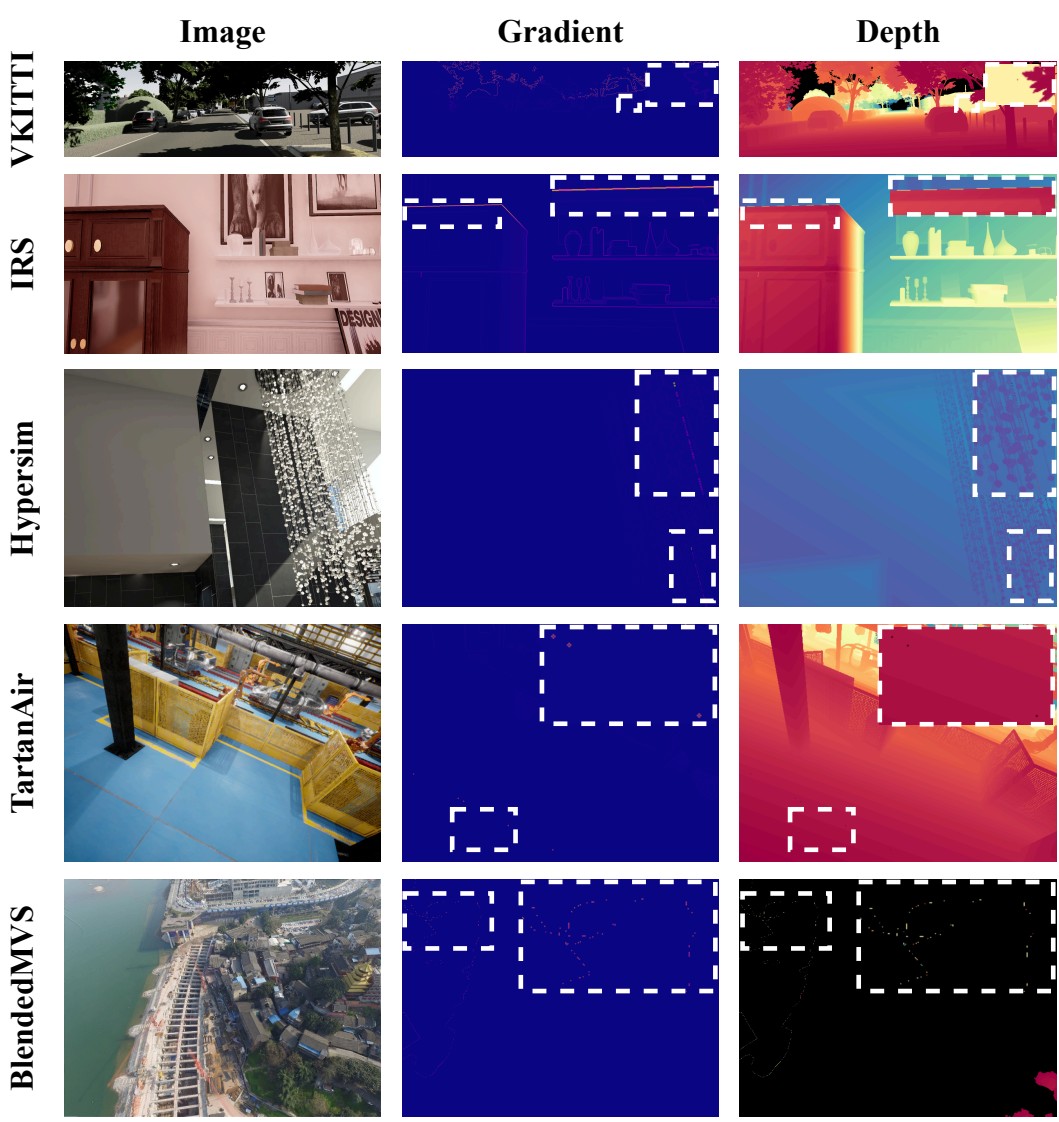

Figure 8: Examples of RGB, gradient, and GT depth from five datasets. The dotted box highlights the noisy area.

