# OpenReview forum: "AnyDepth: Depth Estimation Made Easy"
_ICLR.cc/2026/Conference — ICLR 2026 Conference Withdrawn Submission_

### Official Review · Reviewer_xH2s · 2025-10-19

**Soundness:** 2
**Presentation:** 3
**Contribution:** 2
**Rating:** 4
**Confidence:** 5

**Summary:**

This paper presents a simplified architecture for depth estimation with three major contributions: (1) using frozen DINOv3 as backbone; (2) a lightweight decoder SDT and (3) a data filtering strategy. Experiments show better performance compared with DPT in a constrained setting.

**Strengths:**

- The paper is well writen and it's easy for me to follow.

- Having a more lightweight and effective decoder compared with DPT would be very helpful for the depth community.

- I like the idea of using less data and trying to achieve comparable performance. It's well motivated.

**Weaknesses:**

- It's a bit over-claiming to regard adopting DINOV3 as a major contribution.

- The SDT head should be more carefully ablated to demonstrate better than DPT head. Now, it's only proven to be better than DPT head in a frozen DINOV3 setting. But in most cases, people don't freeze the encoder during training. Will SDT head be better than DPT head when fine-tuning the DINOV3 encoder as well? On the other hand, it would be necessary to use various encoders in experiments and prove that SDT head can generally be better than DPT head with any encoder, DINOV2, ConvNext, etc.

- It would be better to compare the inference time of SDT head and DPT head as well.

- The paper presents three different filtering strategies, but none of them is carefully ablated. What if we only use one/two of these three strategies to do the filtering and train the model? Will we get worse results or not? There is no clue to prove that the combination of these three strategies is better than a simple random sampler.

**Questions:**

Please check the weakness.

---

> ### Author Response · Authors · 2025-11-22
>
> ## Question 1
> We agree with the reviewer that adopting DINOv3 should not be regarded as a major contribution, and we will revise the text to avoid any misunderstanding.
>
> ---
>
> ## Question 2
> To better illustrate the effectiveness of each component, we conducted ablation experiments on five benchmark datasets. As shown in **Table 1**, Filtering, SDE, and Dysample each contribute consistent improvements, and the full model achieves the best performance.
>
> **Table 1. Ablation experiments of AnyDepth-B on five benchmarks. Lower AbsRel and higher δ₁ indicate better performance.**
>
> | Method                      | NYUv2 AbsRel (↓) | NYUv2 δ₁ (↑) | KITTI AbsRel (↓) | KITTI δ₁ (↑) | ETH3D AbsRel (↓) | ETH3D δ₁ (↑) | ScanNet AbsRel (↓) | ScanNet δ₁ (↑) | DIODE AbsRel (↓) | DIODE δ₁ (↑) |
> |-----------------------------|------------------|--------------|------------------|--------------|------------------|--------------|--------------------|----------------|------------------|--------------|
> | w/o Filtering               | 9.5              | 91.1         | 15.4             | 77.3         | 14.0             | 91.2         | 8.3                | 93.5           | 25.0             | 71.1         |
> | Filtering                   | 9.3              | 91.6         | 15.1             | 78.1         | 12.8             | 90.5         | 8.0                | 93.9           | 24.8             | 71.1         |
> | Filtering + SDE             | 8.8              | 92.4         | 14.7             | 79.6         | 11.5             | 91.0         | 7.9                | 94.1           | 24.3             | 71.1         |
> | Filtering + SDE + Dysample  | **7.2**          | **95.0**     | **9.7**          | **90.1**     | **8.0**          | **94.5**     | **6.8**            | **95.6**       | **23.6**         | **72.7**     |
>
> These results demonstrate that each module contributes positively, and combining all components yields a strong cumulative gain.
>
> ---
>
> The purpose of our original experiments was to isolate the contribution of the **decoder architecture**, which is why we evaluated AnyDepth under a frozen-encoder setting. This ensures that improvements can be attributed solely to the decoder rather than encoder optimization.
>
> To directly address the reviewer’s concern, we conducted an additional experiment where **both DPT and AnyDepth are fully fine-tuned together with the same DINOv3 encoder**. As shown in **Table 2**, AnyDepth continues to outperform DPT across all five benchmarks, demonstrating that the advantage of SDT is not restricted to the frozen-encoder scenario.
>
> **Table 2. Comparison between DPT-B and AnyDepth-B when both the encoder (DINOv3) and the decoder are fully fine-tuned. Lower AbsRel and higher δ₁ indicate better performance.**
>
> | Method                  | NYUv2 AbsRel (↓) | NYUv2 δ₁ (↑) | KITTI AbsRel (↓) | KITTI δ₁ (↑) | ETH3D AbsRel (↓) | ETH3D δ₁ (↑) | ScanNet AbsRel (↓) | ScanNet δ₁ (↑) | DIODE AbsRel (↓) | DIODE δ₁ (↑) |
> |-------------------------|------------------|--------------|------------------|--------------|------------------|--------------|--------------------|----------------|------------------|--------------|
> | DPT       | 7.1              | 95.1         | 10.1             | 88.5         | 9.6              | 93.2         | 6.8                | 95.3           | 24.2             | 73.9         |
> | **AnyDepth** | **6.8**          | **95.7**     | **9.0**         | **91.2**     | **7.7**          | **94.7**     | **5.9**            | **96.2**       | **23.1**         | **73.9**     |
>
> These results show that SDT maintains its improvements even when the encoder is fully trainable, confirming that the gains come from the decoder design rather than the frozen-encoder setting.
>
> We conducted an additional experiment using the same backbone, **DINOv2**, for both DPT and AnyDepth. All training settings were kept strictly identical to ensure a fair comparison.
>
> As shown in **Table 2**, **AnyDepth consistently outperforms DPT across all five benchmarks when using the same DINOv2 encoder**.
>
> **Table 2. Comparison between DPT-B and AnyDepth-B (lower AbsRel and higher δ₁ are better).**
>
> | Method   | NYUv2 AbsRel (↓) | NYUv2 δ₁ (↑) | KITTI AbsRel (↓) | KITTI δ₁ (↑) | ETH3D AbsRel (↓) | ETH3D δ₁ (↑) | ScanNet AbsRel (↓) | ScanNet δ₁ (↑) | DIODE AbsRel (↓) | DIODE δ₁ (↑) |
> |----------|------------------|--------------|------------------|--------------|------------------|--------------|--------------------|----------------|------------------|--------------|
> | DPT      | 8.3              | 93.5         | 11.5             | 87.1         | 10.4             | 92.6         | 7.7                | 94.1           | 25.0             | 72.3         |
> | AnyDepth | **7.8**          | **94.7**     | **11.1**         | **87.9**     | **10.1**         | **92.9**     | **7.2**            | **95.1**       | **24.6**         | **73.2**     |

---

> ### Author Response · Authors · 2025-11-22
>
> ## Question 3
> We tested the SDT and DPT headers to provide a comparison of decoder latency only.
>
> All experiments were conducted on an **NVIDIA RTX 4500 Ada Generation** GPU.
>
> **Table 1. Average latency comparison of SDT and DPT decoders across 1000 images (fp32, NVIDIA RTX 4500 Ada architecture).
> Lower latency is better.**
>
> | Input Resolution | DPT (ms) | SDT (ms) | Ratio (DPT / SDT) |
> |------------------|----------|----------|--------------------|
> | 256 × 256        | 13.68 ± 0.11  | 12.27 ± 0.16   | 1.11×             |
> | 512 × 512        | 30.91 ± 0.21  | 30.35 ± 0.29   | 1.02×             |
> | 1024 × 1024      | 131.57 ± 0.27 | 133.29 ± 0.30  | 0.99×             |
>
> The results are consistent with the overall model measurement results: due to its simpler structure, the SDT is faster than the DPT head at low resolutions, while maintaining the same speed at high resolutions.
>
> ---
>
> ## Question 4
> **Table 2. Ablation of different filtering strategies on five benchmarks.
> Lower AbsRel and higher δ₁ indicate better performance.**
>
> | Filtering Strategy         | NYUv2 AbsRel (↓) | NYUv2 δ₁ (↑) | KITTI AbsRel (↓) | KITTI δ₁ (↑) | ETH3D AbsRel (↓) | ETH3D δ₁ (↑) | ScanNet AbsRel (↓) | ScanNet δ₁ (↑) | DIODE AbsRel (↓) | DIODE δ₁ (↑) |
> |----------------------------|------------------|--------------|------------------|--------------|------------------|--------------|--------------------|----------------|------------------|--------------|
> | Depth Distribution Score only    | 7.3              | 94.5         | 9.9             | 89.1         | 8.1              | 93.7         | 7.0                | 95.3           | 23.7             | 72.5         |
> | Gradient Continuity Score only        | 7.4              | 94.1         | 10.1             | 89.2         | 8.6              | 93.0         | 7.0                | 95.4           | 23.8             | 72.5         |
> | **Total Score (ours)**     | **7.2**          | **95.0**     | **9.7**          | **90.1**     | **8.0**          | **94.5**     | **6.8**            | **95.6**       | **23.6**         | **72.7**     |

---

> > ### Comment · Reviewer_xH2s · 2025-11-24
> > **Reply to the Authors**
> >
> > Thanks for the authors providing these detailed replies.
> >
> > From my point of view, this paper presents two seperate methods and sell them in a shared view of simplicity:
> >
> > - (1) Model Level: SDT
> > - (2) Data Level: Filtering data
> >
> > For (1), the major claim or contribution of this paper is to present a better decoder than DPT. But most experiments are conducted in a constrained setting. Though the authors present more experiments using the fine-tuning and dinov2 settings, after reading opinions from other reviewers:
> >
> > Limited comparison with state-of-the-art methods The authors compare their proposed approach only with DPT and Depth Anything v2, which, while relevant, do not represent the current state-of-the-art in monocular depth estimation. To strengthen the empirical validation, comparisons should be made with more recent and competitive methods from xv2A (R1),
> >
> > The scope of the experiments is too limited. The method has not been evaluated in either fine-tuning or metric depth estimation experiments. Experiments only compare to DPT. Although the method is trained on DINOv3 for this study, the approach was first developed in 2021. It would be more intuitive to also compare it to newer, prior-based depth estimation approaches from Mtin (R3),
> >
> > I raised up some other concerns. I agree with the author that these mentioned methods are not directly comparable because they are using other kinds of decoders. However, this is the issue. Recent SoTA tends to adopt various decoders other than DPT, which makes the contribution weaker (even for the metric models).
> >
> > At the same time, DPT is an 'old' paper. A slight improvement over the original DPT can be less persuasive in this case. These experimental results are not enough to draw people's attention to switch from DPT to SDT. I recommand two ways to enhance the quality of this work: (1) use DAv2/3 or VGGT as the baseline, switching from DPT to SDT. fine-tuning them would be fine if the resource is limited. Because they are the most recent SoTA methods involving DPT in their architecture, using them as the base model can be better. (2) develop a SoTA metric model with SDT. Such results can at least encourage people working on metric depth estimation to replace custom designed decoders to SDT.
> >
> > I also agree with the question from reviewer xv2A (R1): How each component **in** SDT takes part in final output performance? is important. Currently, we have no idea about this. I also suggest a layer-wise Flops and # of params comparison to show and present more details.
> >
> > For (2), simlar to reviewer 5CX5 (R2), I feel the improvement is too small.

---

### Official Review · Reviewer_Mtin · 2025-10-29

**Soundness:** 3
**Presentation:** 3
**Contribution:** 2
**Rating:** 4
**Confidence:** 2

**Summary:**

The paper introduces a lightweight framework for zero-shot monocular depth estimation, which combines a frozen DINOv3 encoder with a simple decoder called the Simple Depth Transformer (SDT). Unlike the complex multi-branch design of DPT (related work), the proposed SDT fuses multi-scale tokens once and reconstructs depth through a single upsampling path. This reduces parameters and FLOPs by over 70%, while maintaining comparable accuracy across benchmarks such as NYUv2, KITTI, ScanNet, ETH3D and DIODE. The paper also presents a data-centric filtering strategy that uses depth distribution and gradient continuity metrics to eliminate noisy training samples. This reduces the size of the dataset by one-third without any loss of performance. Experiments demonstrate that AnyDepth achieves a level of accuracy similar to that of larger models such as DPT, but with less computation.

**Strengths:**

The motivation to reduce data requirements of depth estimation and at the same time reduce the parameter count of these models is meaningful and interesting.

The paper shows clear efficiency gains over prior methods while also maintaining/improving overall performance.

The idea is straightforward and the paper is well written. Figures 2 and 3 immediately illustrate the advantages of the method.

It can be considered as the first application of DINOv3 for zero-shot depth estimation.

**Weaknesses:**

There is limited novelty within the data-centric learning metrics. The depth distribution score and gradient continuity score are simple to come up with and lack of some theoretical justification. In the experiments involving these metrics there is no sensitivity analysis or a comparison to other data sampling strategies, e.g. uncertainty aware filtering with techniques such as [1].

The effectiveness of data filtering is incremental. Table 3 shows only a minor benefit from applying the introduced filtering strategy; this does not support the claims.

The scope of the experiments is too limited. The method has not been evaluated in either fine-tuning or metric depth estimation experiments. Experiments only compare to DPT. Although the method is trained on DINOv3 for this study, the approach was first developed in 2021. It would be more intuitive to also compare it to newer, prior-based depth estimation approaches, such as [2, 3, 4].

References:

[1] Hornauer, J., El-Ghoussani, A., & Belagiannis, V. (2025). Revisiting Gradient-Based Uncertainty for Monocular Depth Estimation. *IEEE Transactions on Pattern Analysis and Machine Intelligence*.

[2]Ke, B., Qu, K., Wang, T., Metzger, N., Huang, S., Li, B., ... & Schindler, K. (2025). Marigold: Affordable Adaptation of Diffusion-Based Image Generators for Image Analysis. *arXiv preprint arXiv:2505.09358*.

[3]Garcia, G. M., Abou Zeid, K., Schmidt, C., De Geus, D., Hermans, A., & Leibe, B. (2025, February). Fine-tuning image-conditional diffusion models is easier than you think. In *2025 IEEE/CVF Winter Conference on Applications of Computer Vision (WACV)* (pp. 753-762). IEEE.

[4]Gui, M., Schusterbauer, J., Prestel, U., Ma, P., Kotovenko, D., Grebenkova, O., ... & Ommer, B. (2025, April). DepthFM: Fast Generative Monocular Depth Estimation with Flow Matching. In *Proceedings of the AAAI Conference on Artificial Intelligence* (Vol. 39, No. 3, pp. 3203-3211).

**Questions:**

The low parameter count of the introduced approach is really interesting. Could the authors perhaps add a sensitivity analysis of the fusion strategy?

Could the proposed filtering metrics accidentally remove valuable but rare edge cases (e.g. scenes with strong depth discontinuities)? How can this be avoided?

---

> ### Author Response · Authors · 2025-11-22
>
> **About Weakness 1:**
>
> Our metrics are not designed to be novel depth metrics. Their purpose is to serve as **simple and scalable data filtering heuristics** that remove relatively low-quality synthetic depth maps *before* training. As shown in the paper (pages 12–15), the dataset contains two dominant failure cases: (1) imbalanced or collapsed depth distributions, and (2) noisy or inconsistent gradients. Our two scores directly target these issues and enable efficient large-scale filtering without requiring any model.
>
> Uncertainty-based filtering approaches (e.g., Hornauer et al., 2025) require a **trained depth model**, and each image needs a **forward and backward pass** to compute *prediction uncertainty* rather than *data quality*. These methods are model-dependent, computationally expensive, and **not applicable at the scale of our dataset (584K samples)**. Therefore, they cannot replace our lightweight data-centric filtering strategy.
>
> ---
>
> **About Weakness 2:**
>
> The goal of the filtering stage is not to produce large accuracy gains but to remove clearly corrupted samples and stabilize supervision. To more systematically demonstrate the contribution of filtering, we performed a comprehensive ablation study on five benchmarks. As shown in **Table 1**, Filtering provides consistent gains across multiple datasets, and its benefits become more pronounced when combined with SDE and Dysample.
>
> **Table 1. Ablation experiments of AnyDepth-B on five benchmarks. Lower AbsRel and higher δ₁ indicate better performance.**
>
> | Method                      | NYUv2 AbsRel (↓) | NYUv2 δ₁ (↑) | KITTI AbsRel (↓) | KITTI δ₁ (↑) | ETH3D AbsRel (↓) | ETH3D δ₁ (↑) | ScanNet AbsRel (↓) | ScanNet δ₁ (↑) | DIODE AbsRel (↓) | DIODE δ₁ (↑) |
> |-----------------------------|------------------|--------------|------------------|--------------|------------------|--------------|--------------------|----------------|------------------|--------------|
> | w/o Filtering               | 9.5              | 91.1         | 15.4             | 77.3         | 14.0             | 91.2         | 8.3                | 93.5           | 25.0             | 71.1         |
> | Filtering                   | 9.3              | 91.6         | 15.1             | 78.1         | 12.8             | 90.5         | 8.0                | 93.9           | 24.8             | 71.1         |
> | Filtering + SDE             | 8.8              | 92.4         | 14.7             | 79.6         | 11.5             | 91.0         | 7.9                | 94.1           | 24.3             | 71.1         |
> | Filtering + SDE + Dysample  | **7.2**          | **95.0**     | **9.7**          | **90.1**     | **8.0**          | **94.5**     | **6.8**            | **95.6**       | **23.6**         | **72.7**     |
>
> These results show that Filtering is effective on its own and plays an important cumulative role within the full SDT pipeline. We will update the paper and **replace the original Table 3 with this more comprehensive ablation**, so that the contribution of each component is clearer to readers.
>
> ---
>
> **About Weakness 3:**
>
> We acknowledge the excellent performance of diffusion-based monocular depth estimation methods. However, these methods (e.g., Marigold, DepthFM, and GeoWizard) represent **drastically different model paradigms**, and comparing AnyDepth to these models obscures the architectures and objectives, which are not directly comparable and cannot independently evaluate the effectiveness of our decoder SDT design.
>
> Our goal is not to surpass generative or prior-based methods, but rather to provide the community with a **simple, efficient, and readily usable decoder alternative** based on the widely used DPT framework. Therefore, DPT and Depth-Anything v2 are the most appropriate and fair benchmarks for evaluating AnyDepth.
>
> Nevertheless, we will revise the quantitative comparison section to **group these diffusion/flow matching-based methods into a separate category** to help readers better understand the overview of different model paradigms and clarify the design priorities of SDT.

---

> ### Author Response · Authors · 2025-11-22
>
> ## Question 1
> We conducted a sensitivity analysis. We evaluated several different fusion strategies, including uniform weights, random weights, and complete removal of learnable weights. As shown in Table 1, our learnable soft-limit fusion achieved the best results, proving it to be a robust and effective design choice.
>
> **Table 2. Sensitivity analysis of the SDT fusion strategy on NYUv2. Lower AbsRel and higher δ₁ indicate better performance.**
>
> | Fusion Variant              | AbsRel (↓) | δ₁ (↑) |
> |-----------------------------|------------|--------|
> | Uniform weights             | 7.3        | 94.9   |
> | Random weights              | 7.3        | 94.8   |
> | No learnable weights        | 7.4        | 94.7   |
> | **Softmax-learnable (ours)** | **7.2**    | **95.0** |
>
> ---
> ## Question 2
> Regarding whether the proposed filtering metrics may accidentally remove valuable but rare edge cases, we clarify that the metrics are intentionally designed to avoid this issue.
>
> First, the **Gradient Continuity Score** only penalizes noisy gradients in smooth regions and does **not** penalize genuine geometric discontinuities such as object boundaries, sharp depth jumps, or naturally unbalanced indoor/outdoor compositions. These structures produce coherent, high-magnitude gradients, which are retained by our score.
>
> Second, the **Depth Distribution Score** is applied only to the most extreme portion of samples. In practice, we filter out only **20% of the lowest-quality synthetic depth maps**, as shown in the visualizations in the paper (pages 14–15). These removed samples correspond to clearly corrupted synthetic depth, rather than rare but meaningful scenes.
>
> Finally, after filtering, the remaining dataset (over 360K samples) still covers the full range of scene diversity. This ensures that rare geometric structures are preserved. We will include a brief clarification of this design in the revised version.

---

> > ### Comment · Reviewer_Mtin · 2025-11-25
> >
> > I would like to thank the authors for providing additional information regarding their paper. Having read all the reviews and author responses, I am going to maintain my current score. The evaluation of fine-tuning or metric depth estimation has not been addressed. Furthermore, I agree with the other reviewers that a comparison with the most recent method is necessary.

---

### Official Review · Reviewer_5CX5 · 2025-10-30

**Soundness:** 3
**Presentation:** 3
**Contribution:** 3
**Rating:** 4
**Confidence:** 5

**Summary:**

This paper introduces an efficient framework for zero-shot monocular depth estimation. The authors aim to maintain accuracy while significantly improving efficiency through a three-pronged approach: a powerful but lightweight architecture and a data-centric training strategy. Experiments show that this combined approach achieves accuracy comparable or superior to larger-scale methods while using fewer parameters and less data.

**Strengths:**

The proposed Depth Distribution Score and Gradient Continuity Score are interesting metrics for assessing depth-map quality, but some aspects remain limited.

**Weaknesses:**

(1) The metrics prioritize distribution smoothness over true geometric fidelity. Consequently, a high score does not necessarily correlate with an accurate depth map.
(2)  The selection of hyperparameters—such as the number of bins, weight values, and normalization schemes—appears empirical. The approach lacks theoretical justification or a sensitivity analysis to validate these choices.
(3) The evaluation fails to account for semantic information, which can lead to the unfair penalization of scenes with naturally imbalanced depth distributions (e.g., those dominated by sky or distant objects).
(4) The paper reiterates well-known limitations of data-driven depth estimation without offering novel insights beyond those already discussed in prior works like Depth Anything v1 and v2.
(5) A significant missed opportunity is the failure to integrate these metrics as differentiable loss functions during training, which could have directly improved depth consistency in the models.
(6) The evaluation lacks a thorough breakdown across critical domains (e.g., indoor vs. outdoor, dense vs. sparse LiDAR). Furthermore, the proposed data filtering strategy demonstrates only marginal performance gains.
(7) The central "data-centric" perspective is not convincingly supported by the experimental results, as the method's performance remains comparable to, but does not surpass, existing baselines.

**Questions:**

Please refer to the weaknesses.

---

> ### Author Response · Authors · 2025-11-23
>
> Thank you for the detailed comments from the reviewers. We will now address points (1) through (7) one by one.
>
> (1)
> We acknowledge that our filtering metrics cannot directly guarantee the correctness of the geometry; however, the correctness of the geometry depends on the *synthetic data generation process itself*, which is beyond the control of the model. Our goal is not to re-estimate the geometry, but to detect and remove obviously corrupted synthetic samples. Our distribution- and gradient-based signals are designed precisely for this purpose.
>
> (2)
> Since these metrics must efficiently process over 500,000 synthetic depth maps, we intentionally used a simple and fast data filtering method. At this stage, our primary focus is on stability and scalability, rather than theoretical optimality. To better understand the impact of each component, we conducted additional ablation experiments (as shown below).
>
> **Table 1. Ablation experiments of different filtering strategies on five benchmarks.
> Lower AbsRel and higher δ₁ indicate better performance.**
>
> | Filtering Strategy               | NYUv2 AbsRel (↓) | NYUv2 δ₁ (↑) | KITTI AbsRel (↓) | KITTI δ₁ (↑) | ETH3D AbsRel (↓) | ETH3D δ₁ (↑) | ScanNet AbsRel (↓) | ScanNet δ₁ (↑) | DIODE AbsRel (↓) | DIODE δ₁ (↑) |
> |----------------------------------|------------------|--------------|------------------|--------------|------------------|--------------|--------------------|----------------|------------------|--------------|
> | Depth Distribution Score Only    | 7.3              | 94.5         | 9.9              | 89.1         | 8.1              | 93.7         | 7.0                | 95.3           | 23.7             | 72.5         |
> | Gradient Continuity Score Only   | 7.4              | 94.1         | 10.1             | 89.2         | 8.6              | 93.0         | 7.0                | 95.4           | 23.8             | 72.5         |
> | **Total Score (ours)**           | **7.2**          | **95.0**     | **9.7**          | **90.1**     | **8.0**          | **94.5**     | **6.8**            | **95.6**       | **23.6**         | **72.7**     |
>
> We will explore these hyperparameters more systematically in future work.
>
> (3)
> We clarify here that our metrics target impaired synthesis depth, not semantic patterns. Valid long-range scenes maintain consistent gradients and stable distributions, and therefore are not removed in practice. The focus of filtering is on unreasonable or noisy synthesis depths, rather than rare but valid semantic cases. In scenarios dominated by the sky or distant objects, our visualizations revealed that some samples exhibited dead pixels, as shown in Figure 7 of the original paper. This also interfered with the normal training of the model.
>
> (4)
> DAv1 used large-scale real and synthetic depth data, while DAv2 completely replaced real depth data with synthetic data and demonstrated strong generalization ability.
>
> Our insights differ: **we reduced the amount of synthetic data but improved its quality**, indicating that higher quality synthetic depth data can outperform a larger amount of synthetic depth data in monocular depth estimation. We believe this is a new insight built upon the insights of DAv2.

---

> ### Author Response · Authors · 2025-11-23
>
> (5)
> We agree that this is an interesting direction, but it is not the primary focus of our work. Our intention is to simplify overall monocular depth estimation, focusing on cleaning the dataset itself and avoiding the introduction of new loss terms or complex optimizations. We will propose this as a possible future extension and will continue to conduct research in this area.
>
> ---
>
> (6-7)
> We have now conducted more comprehensive ablation experiments, and the updated results are shown in **Table 2** below. We will replace the original Table 3 in the paper with this expanded version.
>
> **Table 1. Ablation experiments of AnyDepth-B on five benchmarks.
> Lower AbsRel and higher δ₁ indicate better performance.**
>
> | Method                        | NYUv2 AbsRel (↓) | NYUv2 δ₁ (↑) | KITTI AbsRel (↓) | KITTI δ₁ (↑) | ETH3D AbsRel (↓) | ETH3D δ₁ (↑) | ScanNet AbsRel (↓) | ScanNet δ₁ (↑) | DIODE AbsRel (↓) | DIODE δ₁ (↑) |
> |-------------------------------|------------------|--------------|------------------|--------------|------------------|--------------|--------------------|----------------|------------------|--------------|
> | No Filtering                  | 9.5              | 91.1         | 15.4             | 77.3         | 14.0             | 91.2         | 8.3                | 93.5           | 25.0             | 71.1         |
> | Filtering                     | 9.3              | 91.6         | 15.1             | 78.1         | 12.8             | 90.5         | 8.0                | 93.9           | 24.8             | 71.1         |
> | Filtering + SDE               | 8.8              | 92.4         | 14.7             | 79.6         | 11.5             | 91.0         | 7.9                | 94.1           | 24.3             | 71.1         |
> | **Filtering + SDE + Backsampling** | **7.2**          | **95.0**     | **9.7**          | **90.1**     | **8.0**          | **94.5**     | **6.8**            | **95.6**       | **23.6**         | **72.7**     |
>
> Our goal is not to propose a new data-centric paradigm, but rather to demonstrate that a small amount of carefully selected data is more effective than a large amount of unprocessed synthetic data. While the performance improvement is limited, it is more friendly to academic research, and monocular depth estimation is a task that often needs to be deployed on edge devices. Simply increasing the amount of data is not a good point. We believe that a simpler and lighter model, or a simpler and more efficient pipeline, are more meaningful directions.

---

### Official Review · Reviewer_xv2A · 2025-10-30

**Soundness:** 2
**Presentation:** 3
**Contribution:** 2
**Rating:** 4
**Confidence:** 3

**Summary:**

The authors propose AnyDepth, a depth estimation network that, for the first time, integrates DINOv3 into the depth estimation task through their Simple Depth Transformer (SDT) architecture.
They further conduct an analysis of the training datasets and identify imbalances in the ground-truth depth distributions across different datasets. To address this, they filter out unbalanced data pairs to improve training consistency.
Experimental results demonstrate that the proposed method outperforms existing depth estimation architectures, including DPT and Depth Anything v2.

**Strengths:**

1. Lack of comparison to state-of-the-arts
The author compares their proposed method with DPT and Depth-Anything v2, but they are not the best state-of-the-art works and there are better state-of-the-art works that authors needs to compare.

[1] Repurposing Diffusion-Based Image Generators for Monocular Depth Estimation
[2] Metric3d v2: A versatile monocular geometric foundation model for zero-shot metric depth and surface normal estimation
[3] Depth pro: Sharp monocular metric depth in less than a second

2. Ablation study on SDT
The authors contribution is also on the model architecture of SDT including Spatial Detail Enhancer, and Fusion algorithm but there is no ablation study that how much each component effects to the final output.

**Weaknesses:**

1. Limited comparison with state-of-the-art methods
The authors compare their proposed approach only with DPT and Depth Anything v2, which, while relevant, do not represent the current state-of-the-art in monocular depth estimation. To strengthen the empirical validation, comparisons should be made with more recent and competitive methods, such as:

[1] Repurposing Diffusion-Based Image Generators for Monocular Depth Estimation
[2] Metric3D v2: A Versatile Monocular Geometric Foundation Model for Zero-Shot Metric Depth and Surface Normal Estimation
[3] Depth Pro: Sharp Monocular Metric Depth in Less Than a Second

Including these methods would provide a more comprehensive evaluation and clarify the performance gap relative to the latest advancements.

2. Missing ablation study on SDT components
The proposed Simple Depth Transformer (SDT) introduces several architectural contributions, including the Spatial Detail Enhancer and the Fusion algorithm. However, the paper lacks an ablation study demonstrating the individual contribution of each component to the final performance. Providing such an analysis would help justify the design choices and highlight which modules most significantly impact the overall results.

**Questions:**

1. How each component in SDT takes part in final output performance?

2. How much performance would be changed if you change the encoder of AnyDepth as Dinov2?

3. How much performance would be changed if you change the encoder of DPT as Dinov3?

---

> ### Author Response · Authors · 2025-11-20
>
> Thank you to the reviewers for their valuable feedback.
>
> ---
>
> ## Question 1
> To better illustrate the effectiveness of each component, we conducted ablation experiments on five benchmark datasets. As shown in **Table 1**, Filtering, SDE, and Dysample each contribute consistent improvements, and the full model achieves the best performance.
>
> **Table 1. Ablation experiments of AnyDepth-B on five benchmarks. Lower AbsRel and higher δ₁ indicate better performance.**
>
> | Method                      | NYUv2 AbsRel (↓) | NYUv2 δ₁ (↑) | KITTI AbsRel (↓) | KITTI δ₁ (↑) | ETH3D AbsRel (↓) | ETH3D δ₁ (↑) | ScanNet AbsRel (↓) | ScanNet δ₁ (↑) | DIODE AbsRel (↓) | DIODE δ₁ (↑) |
> |-----------------------------|------------------|--------------|------------------|--------------|------------------|--------------|--------------------|----------------|------------------|--------------|
> | w/o Filtering               | 9.5              | 91.1         | 15.4             | 77.3         | 14.0             | 91.2         | 8.3                | 93.5           | 25.0             | 71.1         |
> | Filtering                   | 9.3              | 91.6         | 15.1             | 78.1         | 12.8             | 90.5         | 8.0                | 93.9           | 24.8             | 71.1         |
> | Filtering + SDE             | 8.8              | 92.4         | 14.7             | 79.6         | 11.5             | 91.0         | 7.9                | 94.1           | 24.3             | 71.1         |
> | Filtering + SDE + Dysample  | **7.2**          | **95.0**     | **9.7**          | **90.1**     | **8.0**          | **94.5**     | **6.8**            | **95.6**       | **23.6**         | **72.7**     |
>
> These results demonstrate that each module contributes positively, and combining all components yields a strong cumulative gain.
>
> ---
>
> ## Question 2
> We conducted an additional experiment using the same backbone, **DINOv2**, for both DPT and AnyDepth. All training settings were kept strictly identical to ensure a fair comparison.
>
> As shown in **Table 2**, **AnyDepth consistently outperforms DPT across all five benchmarks when using the same DINOv2 encoder**.
>
>
> **Table 2. Comparison between DPT-B and AnyDepth-B (lower AbsRel and higher δ₁ are better).**
>
> | Method   | NYUv2 AbsRel (↓) | NYUv2 δ₁ (↑) | KITTI AbsRel (↓) | KITTI δ₁ (↑) | ETH3D AbsRel (↓) | ETH3D δ₁ (↑) | ScanNet AbsRel (↓) | ScanNet δ₁ (↑) | DIODE AbsRel (↓) | DIODE δ₁ (↑) |
> |----------|------------------|--------------|------------------|--------------|------------------|--------------|--------------------|----------------|------------------|--------------|
> | DPT      | 8.3              | 93.5         | 11.5             | 87.1         | 10.4             | 92.6         | 7.7                | 94.1           | 25.0             | 72.3         |
> | AnyDepth | **7.8**          | **94.7**     | **11.1**         | **87.9**     | **10.1**         | **92.9**     | **7.2**            | **95.1**       | **24.6**         | **73.2**     |
>
> ---
>
> ## Question 3
> In the original paper’s **Table 2**, DPT already uses **DINOv3** as the encoder. Therefore, the updated results of DPT with DINOv3 are already included in the baseline comparisons and do not require additional experiments.
>
> ---

---

> ### Author Response · Authors · 2025-11-20
>
> **Rebuttal to “Missing comparison with state-of-the-art methods”**
>
> Our work focuses on improving the **decoder architecture (SDT)** in encoder–decoder monocular depth estimation frameworks. The goal is to address specific limitations in DPT, such as the heavy re-assemble module, multiple alignment operations, high complexity, slow inference, and the use of fixed bilinear upsampling that leads to blurred edges and loss of spatial details.
>
> For this reason, we compare with **DPT** and **Depth-Anything v2**, because:
> - They use a **clean and standard encoder–decoder framework**.
> - They do **not** rely on extra geometric constraints, diffusion priors, or multi-task signals.
> - They allow us to **isolate and highlight the contribution of the SDT decoder** in a fair and controlled setting.
> - This comparison setup is **simple, effective, and most directly reflects the advantages introduced by our SDT design**.
> - Moreover, **DPT is currently the most widely adopted decoder in the community**, and many recent depth methods directly use DPT as their default decoder. Improving upon DPT is meaningful, and we hope that **SDT can serve as a new alternative decoder for future work**.
>
> In contrast:
>
> - Diffusion-based depth methods (e.g., Marigold, GeoWizard, Lotus, GenPercept) rely on **large diffusion priors**, which are fundamentally different from encoder–decoder architectures. Therefore, it is not suitable as a baseline for evaluating SDT.
> - Metric3D v2 and DepthPro introduce **additional constraints, complex structures, or metric-depth objectives**, which are beyond the scope of assessing improvements to a standard decoder.
> - Our paper focuses on **affine-invariant depth**, whereas many of these methods target **metric depth**, further complicating direct comparison.
>
> Therefore, DPT and Depth-Anything v2 are the most appropriate baselines to evaluate SDT fairly and directly.

---

### Note · Authors · 2025-11-29

I have read and agree with the venue's withdrawal policy on behalf of myself and my co-authors.